# An Alternative Conceptual Model for the Spent Nuclear Fuel–Water Interaction in Deep Geologic Disposal Conditions

**Barbara Pastina** [1,*] and **Jay A. LaVerne** [2,*]

1. Posiva Oy, Olkiluoto, 27160 Eurajoki, Finland
2. Radiation Laboratory, Department of Physics, University of Notre Dame, Notre Dame, IN 46556, USA
* Correspondence: barbara.pastina@posiva.fi (B.P.); laverne.1@nd.edu (J.A.L.)

**Featured Application: Long-term safety assessment of spent nuclear fuel deep geologic repositories.**

**Abstract:** For the long-term safety assessment of direct disposal of spent nuclear fuel in deep geologic repositories, knowledge on the radionuclide release rate from the $UO_2$ matrix is essential. This work provides a conceptual model to explain the results of leaching experiments involving used nuclear fuel or simulant materials in confirmed reducing conditions. Key elements of this model are: direct effect of radiation from radiolytic species (including defects and excited states) in the solid and in the first water layers in contact with its surface; and excess $H_2$ may be produced due to processes occurring at the surface of the spent fuel and in confined water volumes, which may also play a role in keeping the spent fuel surface in a reduced state. The implication is that the fractional radionuclide release rate used in most long-term safety assessments ($10^{-7}$ year$^{-1}$) is over estimated because it assumes that there is net $UO_2$ oxidation caused by radiolysis, in contrast with the alternative conceptual model presented here. Furthermore, conventional water radiolysis models and radiation chemical yields published in the literature are not directly applicable to a heterogeneous system such as the spent fuel–water interface. Suggestions are provided for future work to develop more reliable models for the long-term safety assessment of spent nuclear fuel disposal.

**Keywords:** spent nuclear fuel; long-term safety assessment; radioactive waste disposal; radiolysis; interfacial processes

## 1. Introduction

The long-term safety of spent nuclear fuel disposal will have a large impact on the future use of nuclear reactors as a major energy source. Deep geologic disposal is currently the leading long-term strategy for the disposal of spent nuclear fuel, either as is (i.e., with no reprocessing, also referred to as "direct disposal") or as high-level waste (after reprocessing). In the case of direct disposal of nuclear fuel in a deep geologic repository, the fuel assemblies are isolated from the surface environment by a series of engineered barriers embedded in the host rock at a depth of (typically) a few hundred meters [1]. A few disposal sites have been selected to date, for example, the Olkiluoto site in Finland and the Forsmark site in Sweden. In both countries, the KBS-3 method (Figure 1) is implemented to ensure safe disposal. The spent fuel assemblies (consisting of a number of rods containing fuel ($UO_2$) pellets) are encapsulated in a canister that is designed to ensure long-term containment in the conditions expected at the disposal site. According to the KBS-3 disposal method, the canister consists of a cast iron insert (inner canister), providing mechanical stability and a copper shell (outer canister) providing corrosion protection. The canister is surrounded by buffer bentonite, a swelling clay providing favorable hydraulic, chemical, and mechanical conditions for the canister. The underground openings are backfilled with more swelling clay or a mixture of swelling clay and crushed rock in order to seal the

man-made connections to the surface. The repository depth (in Finland, the nominal depth is 430 m below sea level) provides isolation from the surface environment and mitigates the likelihood of inadvertent human intrusion due to common activities, such as underground construction and installation of drinking water wells. The hydrogeochemical conditions at repository depth at the selected disposal sites are typically reducing, supporting the longevity of the disposal canisters.

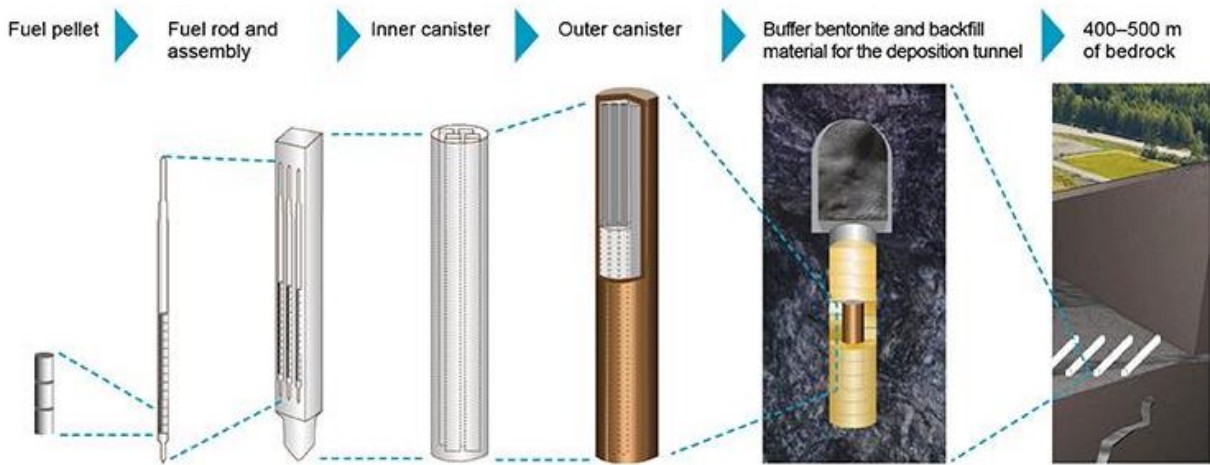

**Figure 1.** Schematic illustration of management and direct disposal of spent nuclear fuel according to the KBS-3 method (Copyright: Posiva Oy).

However, groundwater will eventually penetrate the disposal canisters and come in contact with the spent fuel. Nuclear fuel used in most commercial nuclear power plants consists of uranium dioxide ($UO_2$) enriched with U-235, which is the fissile material. Most radionuclides (e.g., actinides and fission products) produced during the irradiation of nuclear fuel in the reactor are trapped in the $UO_2$ crystalline matrix. The $UO_2$ crystalline matrix is a fluorite-type structure with small distortions and a slight excess stoichiometry ($UO_{2+x}$) due to the effect of irradiation and the formation of fission and activation products. Leaching tests of spent nuclear fuel samples (or materials that are designed to simulate spent nuclear fuel) carried out over several decades show that one of the limiting factors in the release of radionuclides into the environment is the intrinsically low dissolution rate of $UO_2$ [1,2]. Such leaching tests also showed that the release of uranium and other elements seems to be inhibited in the presence of a reducing agent in solution (typically $H_2$ or metallic iron) conditions. The mechanism for such an inhibiting effect is still being debated.

Spent nuclear fuel is an intrinsic source of ionizing radiation (alpha, beta, and gamma) and neutrons, and the intensity of the radiation field depends on the irradiation history and the age of the spent fuel. Radiolysis plays an important role in the $UO_2$ alteration mechanism because the energy deposited within the $UO_2$ matrix in combination with reactive species produced by radiolysis in the water layer at or near the interface will both lead to the formation of reactive species governing the redox conditions at the $UO_2$ interface. Understanding oxidation and reduction processes at the spent fuel–water interface will contribute to improving the conceptual model of radionuclide releases and achieve a more realistic assessment of the environmental effects of spent nuclear fuel disposal.

## 2. Spent Nuclear Fuel in Disposal Conditions

Alpha, beta, and gamma radiation are classified as ionizing radiation, which leads to the cleavage of chemical bonds in the traversed medium in a process called "radiolysis". Neutrons are also emitted within the spent fuel matrix (by fissile radionuclides), but their flux is small and they will interact with water mostly by formation of proton recoils. Radiation (ionizing and not) deposits energy within the spent fuel and any surrounding media such as water. The question as to whether spent fuel self-irradiation affects the

dissolution rates has been pondered for decades. It has been found that the activity of the fuel does have an effect on dissolution rates that depends on the fuel age (inversely proportional to its activity) and on the environmental conditions. These dependencies are more fully discussed below. After about 1000 years, the radioactivity of spent fuel will be dominated by alpha-emitting radionuclides. Given that spent fuel is not expected to be in contact with groundwater for at least several thousands of years [3], even in the presence of an initially defective canister [4], the relevant type of ionizing radiation to be considered for its effect on fuel dissolution is alpha radiation. The effects of alpha radiolysis are also relevant to fuel dissolution processes because of the relatively short range (up to 30 micrometers) from the fuel surface in which the energy is deposited, as discussed below.

When discussing the interaction between ionizing radiation and spent fuel, the heterogeneous nature of the system cannot be neglected (see Figure 2). The system is composed of the spent fuel matrix, the solid–water interface, and (bulk) water. The cladding surrounding the $UO_2$ pellets is also in the immediate surroundings but its role in the spent fuel–water interaction will not be discussed in this paper.

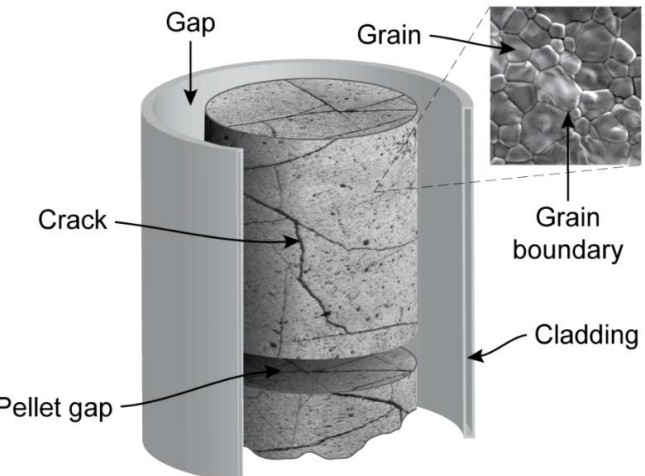

**Figure 2.** The macro- and microstructure of a fuel pellet after its irradiation showing the main features, such as cracks, the surrounding cladding, and the pellet–cladding gap, in addition to the grain boundaries (shown in the inset, at a larger scale). The sizes of the pellet–cladding gap and pellet–pellet gap are exaggerated for clarity [5].

## 3. Observations from Leaching Tests in Reducing Conditions

Table 1 shows a compilation of estimated fractional release rates based on experimentally measured leaching rates obtained in reducing conditions using either alpha-doped $UO_2$ samples (i.e., $UO_2$ samples containing different fractions of short-lived alpha emitters, usually U-233 or Pu-238) or spent fuel samples. The references refer to leaching tests of alpha-doped $UO_2$ samples in reducing conditions, spent fuel, or irradiated MOX samples in reducing conditions, and alpha-doped $UO_2$ samples in Olkiluoto natural or simulated groundwater in the presence of a metallic iron strip. The measured U concentrations in solutions are compared with the solubility limit of the amorphous (more soluble) $UO_2$ form ($2 \times 10^{-9} \leq U < 5 \times 10^{-8}$ M). Table 1 compiles observations by the authors citing negative, fluctuating between positive and negative, or non-measurable release rates in reducing conditions (in the presence of $H_2$ or metallic iron or Fe(II) in solution), even for highly active spent fuel samples and highly alpha-doped $UO_2$ samples. These observations point to an extremely low dissolution rate, if any dissolution at all, in agreement with the conclusion by Shoesmith [2] "*it can be said that based on the above studies there is a strong probability that the corrosion of spent fuel can be avoided either by long term containment in a sealed container or by the reducing influence of $H_2$ in failed container. [ . . . ] Commonly no rate can be measured when $H_2$ is present.*"

From the broader set of results reported in the literature (in the presence and absence of $H_2$ as reducing species), we also know that radiation seems to play an important role in determining whether $H_2$ is effective in suppressing $UO_2$ dissolution. In the absence of radiation, no differences in the oxidation state of the $UO_2(s)$ surface under Ar or $H_2$ were observed, whereas a reduction was reported in the presence $H_2$ when external sources of gamma radiation [6] and alpha radiation [7] were used.

Furthermore, in the absence of radiation (e.g., $UO_2$ leaching tests in $H_2$ saturated solutions), oxygen contamination significantly influences the concentrations of U in solution. In the case of spent fuel and its intrinsic radiation, it appears that the system is not as sensitive to the presence of small amounts of oxygen from air contamination and the concentrations of U in solution, and radionuclide releases (e.g., Cs-137) from spent fuel are not as dramatically affected as may be expected if U(IV) had been oxidized to U(VI) [8–10].

**Table 1.** Summary of release rates in reducing conditions.

| Sample (Reducing Agent) | Activity (Bq g$^{-1}$ UO$_2$) | Observations by the Authors | Reference |
|---|---|---|---|
| PWR segment (H$_2$) 50 MWd kg$^{-1}$ U | $5 \times 10^8$ [a] | Decreasing rates but propose a fractional release rate of $4 \times 10^{-7}$ y$^{-1}$ | Carbol et al. [11] citing Loida et al. [12–14] |
| PWR powder (H$_2$) 43 MWd kg$^{-1}$U | $4.3 \times 10^8$ [a] | $3 \times 10^{-7}$ y$^{-1}$ based on spent fuel powders | Spahiu et al. [15] |
| PWR fragment (Fe) 41 MWd kg$^{-1}$ U | $4.1 \times 10^8$ [a] | Decreasing dissolution rates | Ollila et al. [16] |
| PWR segments (H$_2$) 43 MWd kg$^{-1}$ U | $4.3 \times 10^8$ [a] | Decreasing rates over >1 year | Spahiu et al. [17,18] |
| PWR segment (H$_2$) 43 MWd kg$^{-1}$ U | $4.3 \times 10^8$ | [U] concentrations $10^{-5}$ to $5 \times 10^{-10}$ M practically constant over > 2 years | Ekeroth et al. [9] |
| PWR fragment (H$_2$) 67 MWd kg$^{-1}$ U | $1.2 \times 10^9$ [b] | Decreasing rates followed by steady-state after 502 days | Fors [5] |
| PWR fragments (H$_2$) 40 MWd kg$^{-1}$ U | $4 \times 10^8$ [a] | Inhibited dissolution rates over 3 years | Cera et al. [19] |
| PWR fragments (H$_2$) 65 MWd kg$^{-1}$ U | $1.3 \times 10^9$ [a] | Decreasing U concentrations reaching U solubility limits, inhibition of fuel dissolution | Puranen et al. [20] |
| MOX irradiated (H$_2$) 48 MWd t$^{-1}$ HM | $3 \times 10^9$ [a] | Decreasing U concentrations after 494 days, lower than U solubility limit | Carbol et al. [8] |
| Irradiated MOX (Fe) 47 MWd t$^{-1}$ HM | $3 \times 10^9$ (beta, gamma $2.4 \times 10^{10}$) [b] | No signs of oxidative dissolution | Odorowski 2015 [21] |
| UO$_2$ un-doped (Fe) | $1 \times 10^4$ [b] | No measurable dissolution rate over 1 year | Odorowski 2015 [21] |
| UO$_2$ Pu-doped (Fe) | $3.8 \times 10^9$ [b] | No measurable dissolution rate over 1 year | Odorowski 2015 [21] |
| UO$_2$ Pu-doped (Fe) | $1.8 \times 10^7$ [b] | No measurable dissolution rate over 1 year | Odorowski 2015 [21] |
| Fresh MOX (7%Pu) (Fe) | $1.3 \times 10^9$ (beta, $9.1 \times 10^9$) [b] | No measurable dissolution rate over 1 year | Odorowski 2015 [21] |
| UO$_2$ un-doped (Fe) | $1 \times 10^4$ [b] | $8.3 \times 10^{-8}$ y$^{-1}$ (min) $1.6 \times 10^{-6}$ y$^{-1}$ (max) | Zetterström Evins et al. [22] |
| UO$_2$ U-233-doped 5% (Fe) | $1.6 \times 10^7$ [b] | $7.5 \times 10^{-8}$ y$^{-1}$ (min) $1.1 \times 10^{-6}$ y$^{-1}$ (max) | Zetterström Evins et al. [22] |
| UO$_2$ U-233-doped 10% (Fe) | $3.1 \times 10^7$ [b] | $4.4 \times 10^{-8}$ y$^{-1}$ (min) $1.2 \times 10^{-6}$ y$^{-1}$ (max) | Zetterström Evins et al. [22] |
| UO$_2$ U-233-doped 10% (H$_2$) | $3.3 \times 10^7$ [b] | Decreasing total U concentrations over 328 days. No sign of surface oxidation observed. | Carbol et al. [8] |

[a] Approximation based on Figure 2-1 in Carbol et al. [11] and assuming 10 years after discharge. [b] Reported by the authors.

Therefore, despite existing reports in the literature of oxidative dissolution in both gamma-irradiated $UO_2$ and $UO_2$ doped with alpha emitters, Table 1 shows that there is considerable evidence of the opposite if the conditions are truly reducing, meaning if $H_2$, iron, or other reducing species are dissolved in the leaching solution.

## 4. Current Conceptual Models to Explain the Results in Reducing Conditions

Several authors have proposed explanations for the nil or negative (or alternating between positive and negative) dissolution rates observed in reducing conditions, particularly in the presence of dissolved $H_2$ used as a reducing agent [23–29].

Arguably the most cited conceptual model for the $UO_2$ alteration is the Matrix Alteration Model (MAM), shown in Figure 3, which was initially formulated in the framework of the EU Spent Fuel Stability (SFS) project [30,31]. The MAM conceptual model (and similar ones) is based on the idea of "oxidative dissolution" of spent fuel.

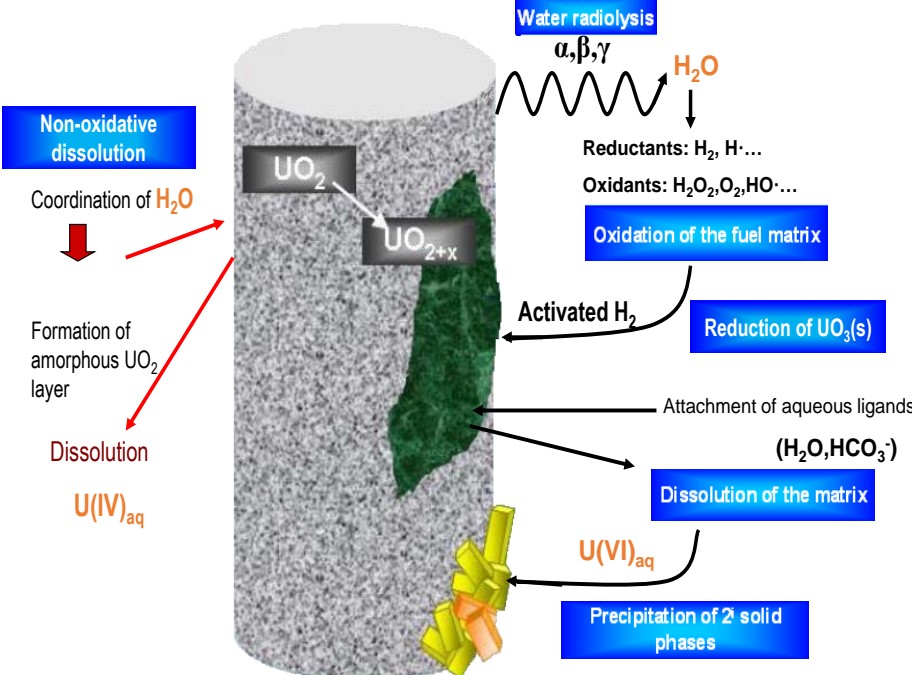

**Figure 3.** Mechanism for $UO_2$ dissolution at macroscopic level according to the Matrix Alteration Model, initially formulated in the framework of the EU Spent Fuel Stability project [31] and further developed by Riba et al. [30].

As shown in Figure 3, water radiolysis (due to alpha, beta, or gamma radiation) produces both reductants and oxidants upon interaction with water. The MAM model always invokes oxidation of U(IV) to U(VI) as the first step of the water-spent fuel interaction.

$H_2O_2$ is often claimed to be the main radiolytic species involved in $UO_2$ oxidation on the basis of either its higher concentration (compared with more reactive radical species such as the hydrated electron and the •OH radical) or faster kinetics compared with other molecular radiolytic species (e.g., $H_2$). Thus, these models imply that the conditions at the spent fuel surface are oxidizing due to the presence of $H_2O_2$.

Furthermore, this conceptual model explains the $H_2$ effect mostly by invoking the activation of dissolved $H_2$ on the surface of $UO_2$ without acknowledging the role of the primary radiolytic products formed in the spent fuel itself and at the water–spent fuel interface. According to the MAM model, the $H_2$ dissolved in bulk water is activated on the surface of the spent nuclear fuel thereby reducing the $UO_3$ in solid form. Aqueous ligands (such as bicarbonate, $HCO_3{}^-$) are able to form complexes with the oxidized U(VI) leading to the dissolution of the $UO_2$ matrix. Secondary solid phases (e.g., schoepite and studtite) can also precipitate on the surface of spent nuclear fuel, if the solubility limit is reached.

Of course, there is always a component of dissolution that is driven by the solubility of the solid (this process is called non-oxidative dissolution in Figure 3). The non-oxidative dissolution is part of the MAM model [31]; it involves the formation of an amorphous $UO_2$ layer leading to the release of U(IV)aq according to the solubility of $UO_2$ (solid phase). The amorphous form of $UO_2$ is more soluble than the crystalline form so the solubility limit

of $UO_2$(amorphous) is typically used to compare with the concentrations of U in solution measured in leaching tests (see Section 3).

The $H_2$ activation mechanism has not been fully explained. What has been clearly demonstrated is that metallic inclusions of Mo, Tc, Pd, Ru, and Rh present in spent fuel (also called epsilon particles) are able to activate $H_2$ and act as catalysts in the reduction of oxidized U(VI). $H_2$ activation on noble metal epsilon particles has been demonstrated electrochemically [24,32] and chemically [26] using SIMFUEL specimens with different levels of simulated burn-up (i.e., different number densities of particles) or added Pd particles, or using radioactive epsilon particles extracted from spent fuel [25]. This work well established that one of the key reactions involved is the catalysis of the reversible dissociation of $H_2$ to ●H radicals on the noble metal epsilon particles and this process galvanically protects $UO_2$ matrix from oxidation.

However, this explanation is not sufficient to justify the lack of a measurable dissolution rate observed in reducing conditions and in the absence of epsilon particles, for example, using alpha-doped $UO_2$ samples (see Table 1). Alternative mechanisms have been proposed in the absence of epsilon particles. For example, Lemmens et al. [33] proposed that the $UO_2$ surface acts as a catalyst in the reduction of oxidized U(VI) back to $UO_2$ or in the recombination of radiolysis products back to water.

The proposed explanations above typically invoke an oxidizing first step due to oxidizing radiolytic species, of which $H_2O_2$ is claimed to be the most important, although recently the role of ●OH radicals has also been mentioned [9,34–36]. The role of the intrinsic ionizing radiation from spent fuel and the effects on the solid and at the spent fuel–water interface are not included in the current conceptual model.

Therefore, although there is consensus on the fact that radiolytic species are involved in the processes occurring at the $UO_2$ surfaces, it is still to be demonstrated whether these species cause oxidation of U(IV) to U(VI) (and possibly involving the intermediate species U(V)) or whether the net balance of oxidizing and reducing radiolytic species in reducing conditions maintains the spent fuel surface in a reduced state so that no oxidizing dissolution occurs.

## 5. Proposed Alternative Conceptual Model

We propose an alternative conceptual model for the observations from leaching tests based on the processes occurring at the early stages of the interaction between ionizing radiation and matter (either spent fuel or water).

### 5.1. Processes Occurring at the Time of Interaction between Ionizing Radiation and Matter

It is well known that the mechanism for water radiolysis proceeds in stages [37,38], as illustrated in Figure 4 by Buxton and Swiatla-Wojcik [39,40]. The first stage is the physical stage, during which time there is an energy transfer between the ionizing radiation and the medium. In the case of pure water, an excited water molecule ($H_2O^*$) and an electron-water cation ($H_2O^+$) pair are formed during the physical stage. The physical stage lasts approximately $10^{-15}$ s.

The second stage is called the physico-chemical stage (lasting approximately from $10^{-15}$ s to $10^{-12}$ s) during which the excited water fragments into ●H and ●OH radicals, in addition to excited O atoms (in triplet or singlet state) and $H_2$. The electrons solvate, and the transfer of a proton from the water cation to a neighboring water molecule yields the ●OH radical. In summary, this stage leads to formation of the primary radiolytic species (●H, ●OH, $H_2$, and hydrated electrons ($e^-_{aq}$).

During the chemical stage (lasting from $10^{-12}$ s to approximately $10^{-6}$ s), the precursor species either escape the non-homogeneous volume where the energy deposition took place or interact with other primary species to form molecular products such as $H_2$, $O_2$, and $H_2O_2$, in addition to the reformation of $H_2O$. Depending on the radiation type, dose and dose rate, and solvents in water, the net balance of these reactions may lead to the recombination of an $H_2O$ molecule with no net radiolysis effect.

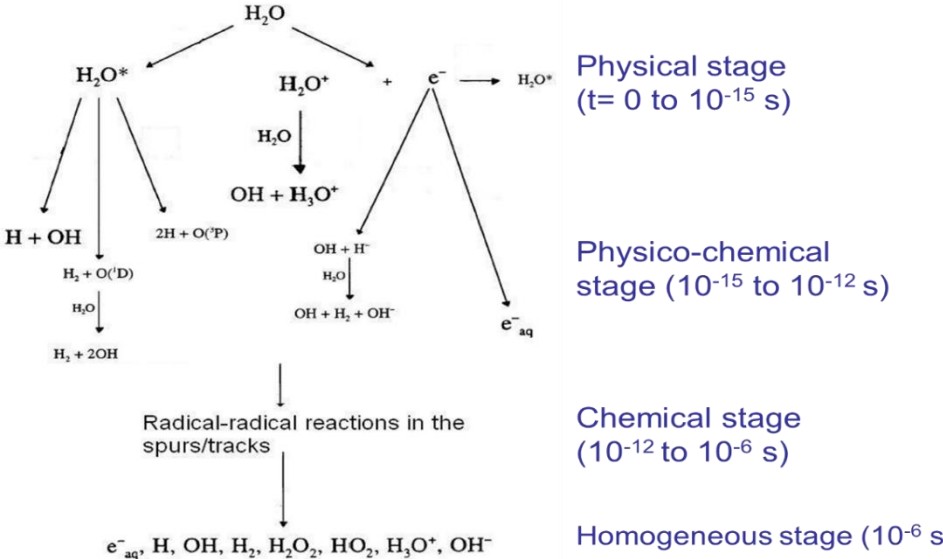

**Figure 4.** Processes occurring upon the interaction of ionizing radiation and a water molecule leading to the production of radiolytic species in solution. Figure prepared based on schemes for the radiolysis of water reported in refs [39,40].

Finally, the homogeneous stage (from $10^{-6}$ s onward) is the stage in which radiolysis products become uniformly distributed in water or aqueous solutions and interact with the environment. In the case of spent nuclear fuel, the sources of radiation are the radionuclides embedded in the solid ($UO_2$) and many of the precursors and primary species may form in the $UO_2$ matrix itself, at its surface or in the environment immediately adjacent to the fuel surface. The energy distribution between the solid and the liquid is also important to determine the nature and fate of such radiolytic species. Alpha radiation in water, for example, will deposit most its energy within 13 microns from the surface and will be completely stopped within 35 microns from the surface [19]. The stopping distance of beta radiation in water ranges from a few mm to tens of mm, depending on the energy. Gamma radiation will travel the farthest from the fuel surface and even penetrate through the cladding surrounding the $UO_2$ pellets and the inner metallic structures of the canister.

The yield of the various radiolytic products depends on the radiation type and energy, and varies as function of time [41]. The primary yield of a particular radiation product, commonly referred to as the G value, is the amount of radiolysis product produced (or destroyed) per unit of absorbed energy. The G value also depends on the linear energy transfer (LET) of the ionizing particle, which is the amount of energy that the particle transfers to the material traversed per unit distance or stopping power ($-dE\,dx^{-1}$). For gamma radiation (low LET), the energy is absorbed in a large volume of water, whereas the short range of alpha particles leads to higher LET. The radiolytic yields (G values) obtained from traditional radiolysis studies in homogeneous systems typically reflect the quantity of species that escape from the spurs and tracks, and are able to interact with solutes during the homogeneous stage.

### 5.2. Processes Occurring in the Bulk of $UO_2$ (Direct Effect of Ionizing Radiation)

Ionizing radiation is known to create electronic defects (such as "holes" formed by the release of an electron from a valence band), or excited states such as polarons or excitons (bound excited states of electrons and holes), and phonons (vibrating atoms in a crystalline structure). Additionally, ionizing radiation can produce Frenkel-type defects created by the displacement of an oxygen atom in the $UO_2$ lattice to form an interstitial-vacancy pair. They are especially common with heavy ions (such as alpha particles) because of the large relative momentum transfer possible in nuclear collisions. The local density of Frenkel defects in a heavy ion track is much greater than that found in gamma radiolysis and

can lead to considerable self-trapping of excitons, the potential precursor to $H_2$ [42,43]. A local increase in the temperature occurs along the high-energy ion path. This temperature increase likely induces an enhanced defect migration leading to defect rearrangement [44].

Defect rearrangement causes point defects to coalesce into a lower energy defect structure (dislocation loops), as observed in self-irradiated $UO_2$, or alpha-doped $UO_2$ or $PuO_2$ samples [44,45]. The local density of Frenkel defects in a heavy ion track is much greater than that found in gamma radiolysis and can lead to considerable self-trapping of excitons, which are bound excited states of electrons and holes, the potential precursor to $H_2$.

Furthermore, Mohun et al. [46] studied the response of $UO_2$, $PuO_2$, and $ThO_2$ to an external source of alpha particles (He ions). Mohun et al. [46] propose that there is a coupling between electronic defects and phonons in $UO_2$ and $PuO_2$, whereas alpha irradiation only causes electronic defects in $ThO_2$. The authors suggest that the alpha-induced irradiation defects may be involved in chemical reactions at the $UO_2$–water interface. In doing so, the $UO_2$ boundary layer acts as a sink and governs the annealing of defects.

Tocino et al. [47] and Corkhill et al. [48] also highlighted the importance of atomistic defects during the dissolution of uranium mixed oxides $(U-Pu)_xO_y$ and cerium oxides $(CeO_2)$, respectively. The authors showed that the high leaching rate of the studied metal oxides was related to the reduced chemical durability of the materials due to the presence of defects in the crystalline lattice, such as oxygen vacancies.

Carbol et al. [11] suggest that the neutralization of the radiolytic oxidants produced near the oxide surface by alpha radiolysis may be due to oxygen vacancies created on the surface of $UO_2(s)$, which likely contributes to the release of hydrogen atoms ($\bullet H$) through the dissociative adsorption of a water molecule. The authors could not rule out that this or other surface processes contribute to the orders of magnitude lower homogeneous radiolytic yields by alpha particles emitted by $UO_2(s)$ surfaces, compared to the yields measured with an external source of alpha radiation.

These findings indicate that species formed due to self-irradiation of $UO_2$ (oxygen vacancies, electronic defects, or excited species) may play a significant role in the chemical reactions occurring at or near the fuel matrix–water interface.

### 5.3. Processes Occurring at the Metal Oxides-Water Interface

The radiolysis of water adsorbed at the surface of metal oxides, including $UO_2$, shows high (~10-fold) radiolytic yields of molecular $H_2$ in adsorbed water compared to that formed by radiolysis of pure water, whereas the yields of $O_2$ were at least an order of magnitude lower for water adsorbed on $UO_2$ or $ZrO_2$ surfaces [42,43,49,50]. The possible mechanism for excess $H_2$ production in the radiolysis of adsorbed water proposed by LaVerne and Tandon [42] (and references therein) involve dissociative electron attachment of low-energy electrons, recombination of electron-hole pairs, and exciton reactions. The negligible production of $O_2$ in these systems may be explained by the decomposition of water bound at the interface, resulting in oxygen species attached to or near the oxide surface.

In 1971, Boehm observed that water associated with oxide surfaces is usually dissociated to form OH groups in the form of hydroxylated metal oxide surfaces ($\equiv$M-OH) [51]. Additional water will be hydrogen-bound to this first chemisorbed layer in the form of physisorbed water layers. The fate of $O_2$ in these systems is unclear. Pulse radiolysis studies suggest that only the reducing species leave the surface to the bulk water [52]. XPS studies of irradiated materials report non-stoichiometric oxygen within the bulk material [53]. The oxygen may be filling up vacancies, but it is most likely breaking up the crystal structure, a process well known in radiation-induced corrosion studies [54].

As mentioned above, self-irradiation of $UO_2$ can cause the formation of oxygen vacancies ($O_v$). Oxygen vacancies ($O_v$) are also known to play an important role in the surface chemistry of metal oxides such as $CeO_2$ and $AmO_2$ (which are fluorite-type

structures often used as $UO_2$ analogues) and $TiO_2$ [55,56]. Hydroxylated species ($\equiv$M-OH) and hydride species ($\equiv$M-H) bonded to the surface can act as oxidizing or reducing species [55,56]. It is also known that oxygen vacancies have the tendency to spontaneously ionize, and thus automatically liberate electron polarons in the lattice and reduce the surrounding metal atoms [57].

First-principles studies of the reaction of water on the crystallographic surfaces of $UO_2$ and $PuO_2$ (and $ThO_2$) help in explaining the excess formation of electrons and $H_2$. Bo et al. [58] propose that $H_2$ is formed from water dissociation. Water dissociation may undergo two pathways in the presence of surface oxygen vacancy on the reduced $UO_2(111)$ surface. One path is characterized by direct combination of two hydrogen atoms of one water molecule, and the other is characterized by dissociation of the first hydrogen atom and its combination with a neighboring surface hydrogen atom.

Wang et al. (2019) [59] propose that excess electrons are formed from radiation-induced oxygen vacancies. The excess electrons lead to the exothermic splitting of $H_2O$ and formation of molecular $H_2$ on $UO_2$ surfaces. This effect depends on the nature of the oxide; for example, for $PuO_2$ the formation of $H_2$ from interfacial water is thermodynamically unfavorable.

Calculations have reported that energies associated with adsorbed and dissociated water are highly dependent on the specific $UO_2$ surface [58,60]. Water adsorbed on a pristine stoichiometric $UO_2$ surface is experimentally found to be a reversible process at room temperature [61]. The evolution of $H_2$ is not observed in temperature programmed desorption studies of these surfaces, indicating that the energy binding the water to the $UO_2$ surface is relatively low [62]. However, water is dissociatively adsorbed on $UO_2$ surfaces with U in a lower oxidation state or from highly defected surfaces. In these latter surfaces, temperature-programmed desorption reveals that the production of $H_2$ is thought to be due to an oxygen vacancy at the surface [62].

The different types of ionizing radiation in spent fuel, because of the different stopping powers or LET, also play an important role in determining the energy distribution and resulting species formed within the $UO_2$ matrix, at the water–spent fuel interface and in bulk water. When a gamma ray, a beta particle, or an alpha particle is released by the radionuclides embedded in spent fuel, part of the energy that is absorbed by the solid forms either defects or excited species that can interact with the water at the interface and beyond. The fraction of the ionizing radiation that is not used in direct effects in the solid causes ionizing events in water within a range that depends on its energy on leaving the surface and LET. Typically, gamma and beta radiation are able to cause ionizing events at a far greater distance than alpha radiation due to their lower LET. When modelling the effect of the intrinsic radiation on water radiolysis, the fraction of the dose absorbed by the $UO_2$ itself should be assessed by taking into account the electron density and the solid/water ratio in the system because it is possible that most of the ionizing radiation energy is absorbed by the solid itself.

One of the more common methods for determination of the energy dissipation through a metal–water interface is the measurement of the yield of the main reducing molecular radiolytic species, $H_2$. The $H_2$ yield has been determined for a wide variety of compounds [42,43,50,53,63,64], including $UO_2$ [43]. In quite a few cases, the production of $H_2$ far exceeds that expected from energy deposited in the water layer alone. Later studies of $UO_2$ found that the radiolytic yield of $H_2$ was about 40 molecules 100 $eV^{-1}$ energy directly absorbed by a few adsorbed water layers, compared to the value of 0.45 molecules/100 eV for bulk water [42]. Clearly, there is a transport of energy or charge from the bulk of $UO_2$ to the interfacial water. The exact precursor to this excess $H_2$ formation is not known but is believed to be due to the transport of excitons [43]. In the next section, we provide a possible mechanism for the production of $H_2$ at the spent fuel–water interface.

### 5.4. Processes Occurring in Confined Water Volumes

The surface of spent fuel is characterized by micro and macro cracks caused by the heavy thermal transients during its service life in the nuclear reactor, as shown in Figure 2. When the fuel comes into contact with water (in scenarios involving canister failure), water penetrates into cracks and gaps, and even reaches grain boundaries. The same processes involved in the interaction between the ionizing radiation and water occur but the solid surface to water volume ratio is different in the radiolysis of confined water.

When studying the radiolysis of confined water, similar high $H_2$ production yields as in the case of radiolysis of metal oxides (see Section 5.3) were observed by Rotureau et al. [65] and LeCaer et al. [66]. The authors explained these effects as due to the lack of scavenging of $H_2$ by the •OH radicals and efficient energy transfer to the pore.

Experiments were performed by Traboulsi et al. [67] on $UO_2$ in alpha irradiated distilled water to study the effect of accumulation of radiolytic $H_2$ in confined water volumes. In those experiments, radiolytic $H_2$ was allowed to escape from the open system but to accumulate in the closed system. In the closed system, the dissolved U concentration was suppressed to about one-third of that observed in the open system due to the accumulation of radiolytic $H_2$. This result shows that, in confined water volumes, radiolytically generated $H_2$ may be sufficient to maintain the surface in reducing conditions and suppress U oxidation without the need for any $H_2$ contribution.

Modelling work by Liu et al. [27,28] also suggests that accumulation of radiolytic $H_2$ in a closed system will completely suppress $UO_2$ dissolution even without an imposed $H_2$ overpressure.

### 5.5. Redox Conditions at the Spent Fuel Surface

As observed in some of the experimental studies discussed in Section 3, we know that radiation plays an important role in explaining the role of $H_2$ in suppressing $UO_2$ dissolution. Although there is consensus on the fact that radiolytic species are involved in the processes occurring at the $UO_2$ surfaces, it is still to be demonstrated which species cause oxidation of U(IV) to U(VI) (possibly involving the intermediate species U(V)) and whether there are species capable of reversing the effect of oxidation so that the net balance of oxidizing and reducing species maintains the spent fuel surface in a reduced state, i.e., no oxidizing dissolution occurs in reducing conditions.

We present below two possible mechanisms producing an excess of reducing species (either •H, $e^-_{aq}$ or $H_2$) that may be responsible for maintaining the spent fuel surface in a reduced state, hence precluding the oxidative dissolution of $UO_2$. The following mechanism is based on the work of Essehli et al. [68] for water adsorbed on $TiO_2$. Radiation first induces the formation of electron ($e^-$) –hole ($h^+$) pairs in $UO_2$, as shown in Equation (1):

$$UO_2 \text{ (intrinsic gamma, beta and alpha irradiation)} \rightarrow h^+ + e^- \tag{1}$$

where $h^+$ is an electronic defect vacancy (see Section 5.2). The charge carriers can combine to give an exciton (hole-electron coupled state on the surface)—Equation (2):

$$h^+ + e^- \rightarrow \text{exciton} \tag{2}$$

It has been suggested that excitons in hydroxylated metals react with the surface water groups ≡U-OH, as shown in Equation (3). The possible formation of hydroxylated groups (≡U-OH) also on the surface of spent nuclear fuel via the formation of oxygen vacancies is discussed in Section 5.3. Lastly, •H atoms combine to form $H_2$—Equation (4):

$$\text{exciton} + \equiv\text{U-OH} \ (+H_2O) \rightarrow \equiv\text{U-O•} + \text{•H} \tag{3}$$

$$\text{•H} + \text{•H} \rightarrow H_2 \tag{4}$$

An additional surface reaction producing $H_2$ can be a hydrogen atom abstraction from a hydroxylated metal surface:

$$\bullet H + \equiv U\text{-}OH \rightarrow \equiv U\text{-}O\bullet + H_2 \tag{5}$$

There is not normally enough adsorbed water to form the hydrated electron, but with nearby bulk water, Equation (3) can take place in the aqueous phase along with the following combination reactions:

$$e^-_{aq} + e^-_{aq} + 2H_2O \rightarrow H_2 + 2OH^- \tag{6}$$

$$e^-_{aq} + \bullet H + H_2O \rightarrow H_2 + \bullet OH \tag{7}$$

Another pathway for $\bullet H$ production may be the electron attachment to $\equiv U–OH$ groups (Equation (8)) followed by dissociation (Equation (9)):

$$e^- + \equiv U\text{-}OH \rightarrow \equiv U\text{-}OH^- \tag{8}$$

$$\equiv U\text{-}OH^- \rightarrow U\text{-}O^- + \bullet H \tag{9}$$

$H_2$ is then formed via $\bullet H$ radical recombination, Equation (4). These mechanisms imply that the conditions at the spent fuel surface are reducing due to the radiation-induced production of excess $H_2$ or other reducing species at the surface. If this is the case, oxidative dissolution does not take place in reducing conditions even without any external input of $H_2$.

The idea that the conditions at the spent fuel surface may be reducing rather than oxidizing has already been proposed by a few authors. King et al. [6], after studying the gamma radiolysis of $UO_2$ in the presence of $H_2$, concluded that hydrogen not only suppresses oxidation due to radiolytic oxidants but also reduces the extent of surface oxidation observed in either Ar or $H_2$ atmospheres in the absence of radiation. The authors propose that this dual effect of $H_2$ is most likely the result of reactions between reducing species and both oxidants in solution and reactive surface sites, the latter possibly located at grain boundaries. The authors could not identify the nature of the reducing species responsible for these effects on the basis of their data, although they invoke the hydrated electron ($e^-_{aq}$) and the $\bullet H$ radical.

Carbol et al. [8] suggests that a combination of alpha-doped $UO_2$, synthetic and anoxic groundwater, and $H_2$ is able to: (a) reduce radiolytic oxidants to water, (b) readily reduce traces of external oxidants entering the system, and (c) stabilize the sample surface as stoichiometric $UO_{2.00}$. The authors propose that $H_2$ dissolved in water is adsorbed on the $UO_2$ surface and acts as electron donor able to reduce oxidizing species formed by radiolysis, and U(V) is potentially formed as a result. Shoesmith [23] proposes the formation of a U(IV)-U(V) site that can then absorb $H_2$. Thereafter, one $\bullet H$ radical is adsorbed and an electron is donated to the surface. Hence, the surface is slightly oxidized but, due to the relatively large amounts of dissolved $H_2$, a back reaction occurs before the next oxidant brings the U(V) to U(VI). The authors propose that such a process would explain the absence of oxidants and oxidized uranium during the long-term leaching tests mentioned in Section 3, and leads to the overall recombination of oxidizing species in solution back to water. The formation of U(V) and subsequent reduction was recently observed at the surface of a thin $UO_2$ film exposed to plasma in vacuum at 400 °C [34,35]. The authors also mention the formation of a surface-bound $\bullet OH$ radical as an intermediate species involved in the oxidation of the $UO_2$ surface [34,35,69].

Hansson et al. [70] also suggested that the effect of dissolved $H_2$ on preventing surface oxidation by alpha radiolysis is apparently a result of the interaction of ionizing radiation with water adsorbed on the surface of actinide oxides, without proposing an exact mechanism but invoking the need of dedicated interfacial radiolysis studies. Hansson et al. [70] recognized that, in addition to standard radiolytic processes, energy, charge, or matter can

be transferred through the interface, and catalytic or steric effects can alter the decomposition or reactivity of adsorbed molecules.

Aqueous phase radiation chemistry can also occur should there be sufficient water near the fuel surface. In that case, homogeneous radiation chemistry will occur with the additional build-up of $H_2$ due to the confined conditions. Under these circumstances, the Allen cycle [71] will lead to the destruction of oxidizing species produced in the aqueous phase:

$$\bullet OH + H_2 \rightarrow H_2O + \bullet H \tag{10}$$

$$\bullet H + H_2O_2 \rightarrow H_2O + \bullet OH \tag{11}$$

Reactions (10) and (11) will cycle until all the oxidizing species are removed as long as there is excess $H_2$.

Excess $H_2$ can also be formed by reactive species formed during the interaction of metal oxides with water, as discussed above. The role of reactive species on metal oxide surfaces in accelerating the dissociation of water and producing $H_2$ in neutral and alkaline solutions has been also proposed in the framework of different studies, such as the development of catalysts for fuel cells [72,73] or metal corrosion [55,74–76]. Betova et al. [74] proposed a model to explain the observed $H_2$ generation in the presence of copper metal and deoxygenated water involving surface sites available for adsorption of hydroxide intermediates acting as catalysts in the splitting of adsorbed water molecules.

According to Li et al. [55,56], in addition to surface hydroxyl species (M≡OH), various reactive species are formed on metal oxide surfaces in contact with water and in the presence of a reducing agent, such as $H_2$, such as hydride species (M≡H) and hydrated protons (M≡H$_3$O$^+$). These M≡OH and M≡H species are formed on the surface and in the bulk, accompanied by oxidation and reduction reactions of the metal [55,56]. According to the authors, oxygen vacancies have a strong influence on both the stability and reactivity of these species. Hydroxyl groups, together with hydride species and hydrated protons, can explain the formation of $H_2$ from the interaction of a metal oxide and water via the formation of H-bonding networks on the surface of the metal oxide, based on previous work [75,76]. The photocatalytic properties of a Pt-TiO$_2$ surface in reducing water to $H_2$ can be explained by such H-bonding networks, which are formed by hydroxyl groups on solid surfaces [55,56]. TiO$_2$ itself exhibits structural hydroxyl groups and limited surface oxygen vacancies where $H_2O$ dissociates to form hydroxyl groups. Moreover, the loading of Pt co-catalysts on TiO$_2$ introduces a Pt–TiO$_2$ interface capable of efficiently dissociating $H_2O$ to form hydroxyl groups, which greatly facilitates the formation of H-bonding networks. Epsilon particles that can be found in spent fuel contain d-metals (Pd, Mo, Tc, Ru, and Rh), which may play a similar catalytic role as that of Pt on a TiO$_2$ surface in dissociating water with the net production of $H_2$.

The alternative conceptual model presented in this paper is shown in Figure 5. The novelty consists in including the interaction of ionizing radiation with the UO$_2$ matrix involving the formation of both oxidizing and reducing species (e.g., hydroxylated species, hydrides, and hydrated protons), which may explain the depletion of the oxidizing radiolysis products and a possible net production of $H_2$ at the surface. These species may be responsible for maintaining the spent fuel surface in a reduced state even without any external input of $H_2$.

In summary, the main source of radiation over long periods will be alpha radiation and its short range will lead to radiolysis at or near the surface. This region is most critical for dissolution. The arguments apply also for confined water in the presence of an intrinsic source of radiation, such as the water that is trapped in cracks at the spent fuel surface and even water that has penetrated in the grain boundaries.

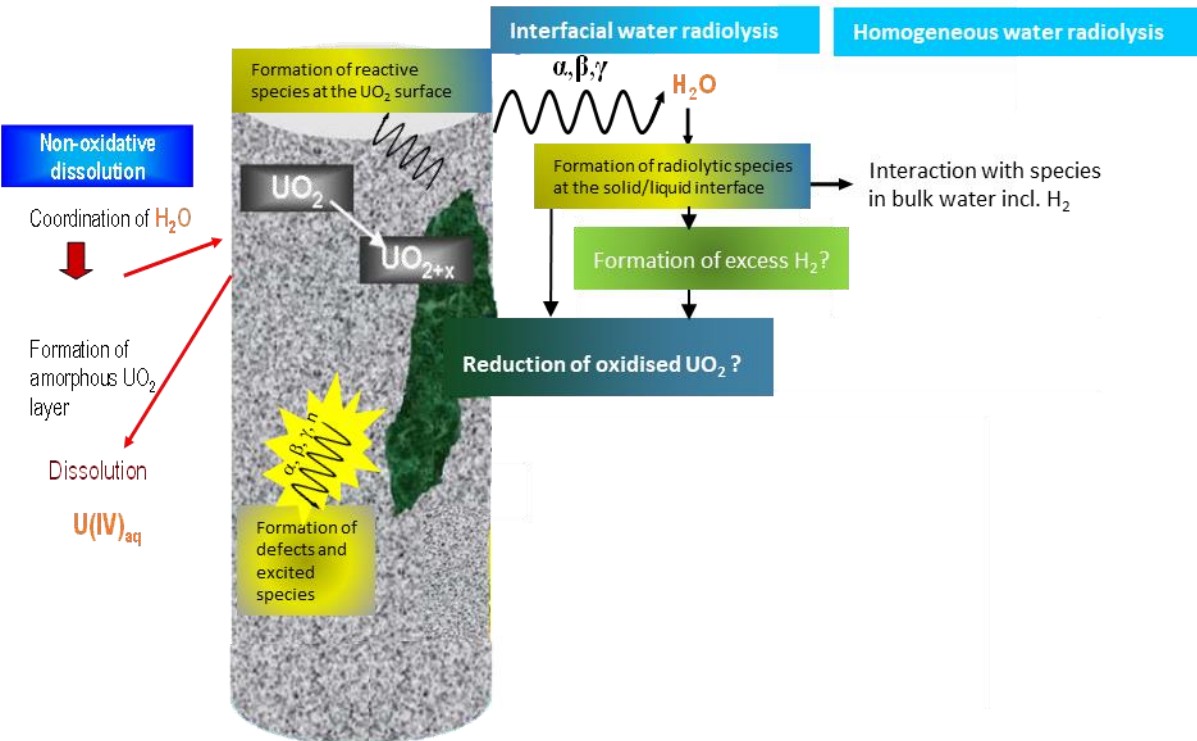

**Figure 5.** Alternative conceptual model for spent fuel–water interaction, modified from Riba et al. [30].

## 6. Outlook on Developments of the Conceptual Model for Spent Fuel–Water Interaction

The role of the direct effect of the different types of radiation in spent fuel and the interaction with the $UO_2$ matrix and with the water at the spent fuel interfaces is just starting to be acknowledged. Although there is consensus on the fact that radiolytic species are involved in the processes occurring at $UO_2$ surfaces, it is still to be demonstrated which species cause oxidation of U(IV) to U(VI) (possibly involving the intermediate species U(V)) and whether there are species capable of reversing the effect of oxidation so that no net $UO_2$ oxidation occurs.

Unfortunately, there is no perfect analogue to study the releases of "aged" spent fuel. Spent fuel samples currently available have the right physical and chemical properties but still contain a significant amount of beta and gamma radiation, being only a few decades old. Alpha-doped $UO_2$ samples simulate correctly the level of activity of "aged" spent fuel but do not have the right physical and chemical properties. Chemical analogues, such as $CeO_2$, have the same crystalline (fluorite) structure as $UO_2$ but no radiation. SIMFUEL, which is synthetic $UO_2$ containing d-group noble-metal particles, similar to those formed in spent nuclear fuel, also does not emit radiation. All these materials together provide useful insights on the dissolution rate of $UO_2$, but none individually.

Both Tocino et al. [32] and Mohun et al. [46] showed that metal oxides which have the same crystalline structure can respond differently to radiation (and/or interact differently with water). Therefore, care should also be taken when comparing $UO_2$ crystalline analogues and drawing conclusions on the energy–water interaction mechanism.

Similarly, water radiolysis studies using external sources of ionizing radiation (typically alpha or gamma or accelerated electrons) in the absence of a solid–liquid interface cannot be used to simulate the effect of intrinsic (or self-) irradiation on spent fuel because typically they address the homogeneous stage of water radiolysis and not the interaction between the species formed due to the direct effect of radiation in the solid and those formed at the solid–water interface. For example, Pastina and LaVerne [77,78] investigated the decomposition of an initial $H_2O_2$ solution ($5 \times 10^{-5}$ mol L$^{-1}$) in bulk water under

external sources of pure gamma and alpha radiation. In the case of an external source of gamma irradiation, complete $H_2O_2$ consumption was observed, confirming the effect of radiolytic species and dissolved $H_2$ in reducing oxidizing species such as $H_2O_2$. However, under an external source of alpha radiation, even if the solution was saturated with $H_2$ (0.8 mM), the initial $H_2O_2$ could not be reduced. In this case, the explanation is that the escape yields of primary radiolytic products from the tracks formed by an external alpha source are too low to interact with solutes in bulk water, such as $H_2$ and $H_2O_2$. In order to interact with the radical species in the alpha particle tracks, much higher concentrations of radical scavengers would be needed, which was not the case in this particular experiment by Pastina and LaVerne [78]. Alternatively, during bulk water radiolysis using a mixed source of gamma and alpha radiation at 25 °C (where the alpha particles were produced in situ by neutron capture in a solution containing $B^{10}$), even modest levels of dissolved $H_2$ (0.8 mM) can suppress the formation of molecular radiolytic oxidants [79].

The different responses for the alpha radiolysis are mainly due to the energy distribution of the alpha particles. In external radiolysis, much of the energy loss occurs at the higher energies and lower LET, so radicals can escape the track and interact with oxidizing species. By comparison, in situ radiolysis results in a substantial portion of the alpha particles being attenuated in energy before reaching the surface. The shift in average LET to lower values reduces the escape yields of radical species. Thus, radiolysis studies in bulk water can only provide limited information on the processes occurring at the spent fuel surface.

It can be argued that even the concept of radiolytic yield does not apply to interfacial water radiolysis. Radiolytic yields (G values) obtained from traditional radiation chemistry experiments cannot be applied in the model of spent fuel dissolution because they apply to the homogeneous phase of water radiolysis, whereas, in this case, the yields of radiolytic species formed during the physical and physical-chemistry stage are of relevance.

In summary, the spent fuel–water interface may play an active role in the radiolysis processes, and the primary yields of radiolytic products may be very different from the G values obtained from traditional radiolysis studies in homogeneous systems. Hence care should be used when applying such G values to model spent fuel dissolution as a function of the radiation dose.

To improve the understanding of the interaction between the spent fuel surface and water, the effect of the intrinsic ionizing radiation on the $UO_2$ crystalline matrix, the confined water volumes present at the spent fuel surface, and the layers of water immediately adjacent to the spent fuel interface should be further investigated. The formation of both oxidizing and reducing species at the surface of $UO_2$ and at the $UO_2$–water interface leads to a large diversity of reaction pathways to be taken into account for better understanding of reactivity of spent fuel in reducing conditions.

The following are examples for potential future studies that could shed further light on the processes occurring at the spent nuclear fuel–water interface:

- Comparison of the effects of alpha particles produced in situ (for example, by using neutron capture on soluble compounds containing B-10) to those of external alpha sources;
- Further exploration of the field of radiolysis in confined water volumes and of water at the surface of metal oxides, in particular, fluorite-type solids;
- Investigation of whether excess $H_2$ can be produced at the spent nuclear fuel–water interface;
- Modelling using Monte Carlo based techniques exploring physico-chemical processes at the spent fuel–water interface.

## 7. Conclusions

This work provides an alternative conceptual model to explain the leaching test results involving actual spent nuclear fuel or simulant materials in (confirmed) reducing conditions, which are relevant for deep geologic disposal.

The key findings are the following: the direct effect of radiation forms radiolytic species (including defects and excited states) in the solid and in the first water layers in contact with its surface; excess $H_2$ may be produced due to processes occurring at the surface of spent fuel and in confined water volumes involving such species; such species may also play a role in keeping the spent fuel surface in a reduced state.

Conventional water radiolysis models cannot be used directly to estimate the radionuclide release rates because models and G values are designed for bulk (homogeneous) chemistry in the absence of a solid–liquid interface. The implication of the alternative conceptual model proposed in this work is that the long-term fractional dissolution rate ($10^{-7}$ $y^{-1}$) used in most long-term safety assessments is overly estimated because this rate assumes that there is a net $UO_2$ oxidation rate caused by radiolysis, in contrast with the alternative conceptual model presented here. (Note that the source term for radionuclide release and transport models used in the long-term safety assessment typically involves a long-term fractional dissolution rate for radionuclides embedded in $UO_2$ and an "instantly" released fraction of the radionuclide inventory to take into account the fraction of radionuclides that are typically released shortly after the spent fuel comes in contact with water. Instantly released radionuclides, such as Cs-137, Cl-36, C-14, and I-129, can be found in locations that are easily accessible to water, such as fractures, gaps, or grain boundaries.)

Because of uncertainties in experiments, in addition to the evolution of the surface area of the fuel in contact with water and long-term self-irradiation effects, adopting a pessimistic long-term fractional dissolution rate and maintaining it at a constant value throughout the assessment period (typically 1 M years) is a cautious approach, at present. This approach is currently used in the source term for long-term safety assessments by spent fuel management organizations, such as Posiva Oy [80], SKB [81], Nagra [82], and NWMO [83].

This paper also provides suggestions for future work that will yield more information on the interfacial water radiolysis occurring at the spent fuel surface. If it can be proven that the conditions at the spent fuel surface are indeed reducing due to the effect of radiolytic species formed in the spent fuel itself or in the close proximity of the surface, then it will be possible to develop more reliable models for the long-term safety assessment of nuclear fuel disposal.

**Author Contributions:** Conceptualization; writing—original draft preparation, writing—review and editing: B.P. and J.A.L. All authors have read and agreed to the published version of the manuscript.

**Funding:** This research received no external funding.

**Institutional Review Board Statement:** Not applicable.

**Informed Consent Statement:** Not applicable.

**Acknowledgments:** Posiva Oy's contribution to this work (B.P.'s time and publication fees) is acknowledged. JAL acknowledges the Notre Dame Radiation Laboratory, which is supported by DOE BES Grant Number DE-FC02-04ER15533. This contribution is NDRL-5332 from the Notre Dame Radiation Laboratory.

**Conflicts of Interest:** The authors declare that they have no known competing financial or personal relationships that could have appeared to influence the work reported in this paper. B.P. is a staff member of a spent fuel management organization (Posiva Oy) which may be perceived as a non-financial potential conflict of interest.

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
