# Peer review of "An Alternative Conceptual Model for the Spent Nuclear Fuel–Water Interaction in Deep Geologic Disposal Conditions"

_applsci, doi:10.3390/app11188566_

Round 1
Reviewer 1 Report
My review is attached

Author Response
Reviewer #1
Point 1: This is an interesting paper which addresses important issues regarding the disposal of spent nuclear fuel. I recommend publication once the following points are considered. The authors present a convincing argument that the results from radiolytic experiments on bulk water are not directly applicable to the interpretation of behaviour at oxide/water interfaces. However, the paper contains a number of inconsistencies and obscure interpretations.
For example, it is unclear whether they are making an argument that irradiation directly produces reducing conditions or that this is the case only in the presence of a small excess of hydrogen.
Some sentences suggest the first statement is their opinion (e.g., page 11 (407-409).
Response: Both arguments are correct. One would expect that under the anaerobic conditions of deep repositories the amount of reducing species would be greater than the oxidizing species with alpha particles. Studies at with different radiations have shown that the yield of the main oxidizing species H2O2 goes down with increasing stopping power of the radiation. This results in a slight over supply of H2 and the Allen cycle of reactions 10 and 11 reduces the oxidizing species. The statement on pages 407-409 refers to the surface of the fuel where H2 gen is being produced in excess. The latter phenomenon is the main focus of the paper.
The following text has been added to page 12 to clarify “These mechanisms imply that the conditions at the spent fuel surface are reducing due to the radiation-induced production of excess H2 or other reducing species at the surface. If this is the case, oxidative dissolution does not take place in reducing conditions even without any external input of H2.”
Point 2: There is a considerable literature demonstrating that both gamma irradiated UO2 and UO2-doped with alpha emitters experience oxidative dissolution.
Response: The reviewer is correct. However, Table 1 shows that there is a considerable evidence of the opposite if the conditions are truly reducing. For many papers in the literature reporting oxidation of U(IV) to U(VI), it is unclear whether the conditions were actually reducing. Simply operating in an autoclave does not guarantee that the conditions are reducing, they could be simply anoxic. For conditions to be reducing either H2 or Fe or another reducing species should be present in solution.
The following text has been added to page 4 “Therefore, in spite of existing literature reporting oxidative dissolution in both gamma-irradiated UO2 and UO2-doped with alpha emitters, Table 1 shows that there is a considerable evidence of the opposite if the conditions are truly reducing, meaning if H2, iron or other reducing species are dissolved in the leaching solution.”
Point 3: It is also unclear whether their arguments apply to spent fuel exposed to bulk water or only to situations where the amount of available water is confined to a few surface layers. This should be made more explicitly clear.
Response: We agree that the discussion of confined water is a bit disconnected from the rest. As discussed in the text, the main source of radiation at long times will be alpha radiation and its short range will lead to radiolysis at or near to the surface. This region is most critical for dissolution. The arguments apply also for confined water in the presence of an intrinsic source of radiation, such as the water that is trapped in cracks at the spent fuel surface and even water that has penetrated in the grain boundaries.
The following text has been added to page 13 “In summary, the main source of radiation at long times will be alpha radiation and its short range will lead to radiolysis at or near to the surface. This region is most critical for dissolution. The arguments apply also for confined water in the presence of an intrinsic source of radiation, such as the water that is trapped in cracks at the spent fuel surface and even water that has penetrated in the grain boundaries. “
Point 4: A clear argument is made that irradiation influences not only the solution redox conditions but also the near-surface structure of the UO2. While such arguments have been made previously for the alpha irradiation of spent fuel and UO2 this is the first time they have been coherently discussed and extensively compared to observations on other oxides, as well as extended to the influence of gamma radiation. This manuscript would have benefitted from separate discussions of alpha irradiation, when the influence of oxygen vacancies appears to be clear, and gamma radiation, when it is not.
Response: Oxygen vacancies will affect the bulk processes due to alpha radiolysis and they may alter the transport of energy to the surface. There is insufficient data on the latter process to elaborate. This manuscript concentrates on the radiation chemistry at the solid-water interface and concerned with energy deposition at that interface. The authors are not sure what they can add to the discussion on oxygen vacancies as they affect the surface radiation chemistry.
Point 5: In the discussion on page 9, no credible argument is presented to explain the fate of the O species produced when radiation leads to hydrogen generation. For alpha irradiation it can be argued it is incorporated in the UO2 by eliminating oxygen vacancies. For gamma irradiation the following obscure explanation is offered, “The negligible production of O2 in these systems could be explained by the decomposition of water bound at the interface resulting in oxygen species attached to the surface or near the surface.” This statement avoids the need to explain the fate of the reactive radical species (O•, OH•) produced in reactions 3, 5, 7. Where does the oxygen finally go?
Response: No one knows exactly where the oxygen goes. It does not appear to leave in the form of O2. Pulse radiolysis studies suggest that only the reducing species leave the surface to the bulk water. The references cited state that XPS studies of irradiated materials find non-stoichiometric oxygen within the bulk material. The oxygen could be filling up vacancies, but it is most likely breaking up the crystal structure, a process well known in radiation induced corrosion studies.
The following text has been added to page 9 “The fate of oxygen in these systems is unclear. Pulse radiolysis studies suggest that only the reducing species leave the surface to the bulk water [52]. XPS studies of irradiated materials report non-stoichiometric oxygen within the bulk material [53]. The oxygen could be filling up vacancies, but it is most likely breaking up the crystal structure, a process well known in radiation-induced corrosion studies [54].”
Point 6: A number of smaller issues should be addressed.
- 6a: Page 5 (140). It is not clear what the term “most diffused conceptual model” means.
Response: This wording was changed to "most cited conceptual model" and a reference given to the EU SFS project , which is the project that started looking into the conceptual model of the dissolution of spent fuel. - 6b: Page 8 (282). What is meant by “unphysically high”?
Response: The “unphysical” has been eliminated. The statement is that high H2 yields are observed as compared to bulk water. - 6c: References 31 and 49 are not the correct references. I have not checked many references, but the authors need to check them all.
Response: The two references have been corrected. All references have been rechecked.
Reviewer 2 Report
The authors of applsci-1381290 present an interesting and well-designed short review dedicated to the conceptual model for the spent nuclear fuel interaction in deep disposal conditions. The manuscript corresponds to the subject of the journal. The figures and tables included in the work help in explanations of the concepts in the paper.
However, some elements must be improved. In this regard, my major comments include:
- Follow the IUPAC guideline on formatting units, i.e. avoid x/y and use x y^-1 consistently throughout the manuscript.
- Use subscript in noting the chemical substances, i.e.UO2, H Please standardize this throughout the manuscript.
- Figure 1. has not been mentioned and described in the manuscript text.
- Please standardize the way of writing species. For example, once you write them using "-" and once "–" (≡M-H; ≡U—OH).
- You refer to Wang et al. (2019) but this work is not in the Reference list.
- Avoid overcitation, for example [48; 49; 42; 58; 43; 59-62].
Author Response
Reviewer #2
Specific editorial comments
- Follow the IUPAC guideline on formatting units, i.e. avoid x/y and use x y^-1 consistently throughout the manuscript.
Response: The appropriate format has been adopted.
- Use subscript in noting the chemical substances, i.e.UO2, H Please standardize this throughout the manuscript.
Response: The nomenclature has been standardized. - Figure 1. has not been mentioned and described in the manuscript text.
Response: Figure 1 is now mentions in the appropriate place and a description added on page 1 -2. - Please standardize the way of writing species. For example, once you write them using "-" and once "–" (≡M-H; ≡U—OH).
Response: The dashes for species has been standardized. - You refer to Wang et al. (2019) but this work is not in the Reference list.
Response: The reference has been added
- Avoid overcitation, for example [48; 49; 42; 58; 43; 59-62].
Response: This citation has been reduced in number.